# CD3^+^, CD8^+^, CD4^+^ and FOXP3^+^ T Cells in the Immune Microenvironment of Small Bowel Neuroendocrine Tumors

**DOI:** 10.3390/diseases9020042

**Published:** 2021-06-11

**Authors:** Niko Hiltunen, Juha P. Väyrynen, Jan Böhm, Olli Helminen

**Affiliations:** 1Cancer and Translational Medicine Research Unit, Medical Research Center, University of Oulu, Oulu University Hospital, 90220 Oulu, Finland; juha.vayrynen@oulu.fi (J.P.V.); olli.helminen@oulu.fi (O.H.); 2Department of Pathology, Central Finland Central Hospital, 40620 Jyväskylä, Finland; jan.bohm@ksshp.fi; 3Surgery Research Unit, Medical Research Center, University of Oulu, Oulu University Hospital, 90220 Oulu, Finland

**Keywords:** neuroendocrine tumor, small bowel neuroendocrine tumor, gastric neuroendocrine tumor, CD3^+^, CD8^+^, CD4 ^+^, FOXP3^+^, prognosis, inflammation

## Abstract

The role of inflammation in neuroendocrine tumors is poorly known. The purpose of this study was to characterize the densities of CD3^+^, CD8^+^, CD4^+^ and FOXP3^+^ T cells in small bowel neuroendocrine tumors (SB-NETs), SB-NET lymph node metastases and gastric neuroendocrine tumors (G-NETs) to assess the prognostic role of immune cell infiltrates in SB-NETs. The final cohort included 113 SB-NETs, 75 SB-NET lymph node metastases and 19 G-NETs from two Finnish hospitals. CD3^+^- and CD8^+^-based immune cell score (ICS), and other T cell densities were evaluated. Survival analyses of SB-NETs and SB-NET lymph node metastases were performed with the Kaplan-Meier method and Cox regression adjusted for confounders. The primary outcome was disease-specific survival (DSS). No significant difference in DSS was seen between low and high ICS groups in SB-NETs at 5 years (92.6% vs. 87.8%) or 10 years (53.8% vs. 79.4%), *p* = 0.507, or in SB-NET lymph node metastases at 5 years (88.9% vs. 90.4%) or 10 years (71.1% vs. 59.8%), *p* = 0.466. Individual densities of the examined T cell types showed no correlation with prognosis either. SB-NETs and lymph node metastases had similar inflammatory cell profiles, whereas in G-NETs CD3^+^ and CD8^+^ T cells were particularly more abundant. In SB-NETs, ICS or T cell densities showed no correlation with prognosis.

## 1. Introduction

Neuroendocrine tumors (NETs) are relatively rare neoplasms, developing in virtually any organ system from cells sharing a neural-endocrine phenotype [1]. Clinical presentation and prognosis of NETs vary significantly between different anatomic sites [2]. Among the most common NETs are gastroenteropancreatic NETs (GEP-NETs), which develop from the enteroendocrine cells of the gastrointestinal tract and pancreas [3]. Small bowel neuroendocrine tumors (SB-NETs) and gastric neuroendocrine tumors (G-NETs) are two separate entities within GEP-NETs.

Based on data from the National Cancer Institute’s Surveillance, Epidemiology, and End Results (SEER) Program, the annual incidence rate of all GEP-NETs is 3.56 per 100,000 persons, including 1.05 per 100,000 persons for SB-NETs [3], while the annual incidence of G-NETs is somewhere between 0.08 and 0.45 per 100,000 persons [3,4]. Incidence of all NETs has increased 6.4-fold over recent decades. The biggest increase has been seen in G-NETs where incidence has grown 15-fold [3]. This increase is at least partly due to improved diagnostic methods in imaging and endoscopy.

Tumor-infiltrating lymphocytes (TILs) are important effectors in antitumor immunity. High CD3^+^ and CD8^+^ T cell densities have been shown to be associated with good prognosis in various cancers [5,6]. Immunoscore is a prognostic parameter based on quantifying the in situ immune reaction by measuring CD3^+^ and CD8^+^ T cell densities at the tumor center and at the invasive margin. Immunoscore was first described in colorectal cancer, where its prognostic significance was later validated internationally [7,8,9]. The prognostic value of Immunoscore has been examined in other cancers, including pancreatic NETs (pNETs), where the results suggest that a nomogram encompassing the Immunoscore system for pNETs (ISpnet) and patient-specific clinicopathological characteristics could effectively predict recurrence-free survival [10,11].

CD4 is a co-receptor of the T cell receptor, expressed in helper T cells and facilitating the interaction with antigen-presenting cells. Helper T cells may, in turn, activate CD8^+^ cytotoxic T cells and support cellular antitumor immunity [12]. Regulatory T cells, distinguished by their FOXP3 expression, contribute to the downregulation of effector T cell responses, potentially promoting tumor immune escape. Accordingly, high densities of FOXP3^+^ T cells have been associated with poor prognosis in many cancers [5,6].

Contrary to many other tumor types, relatively few studies have evaluated the SB-NET immune microenvironment. We see that the trend to evaluate disease microenvironments is growing significantly [13,14]. Therefore, the primary aim of this study was to test the prognostic significance of immune cell score (ICS) based on CD3^+^ and CD8^+^ T cells in a retrospective, consecutive series of SB-NETs from two institutions in Northern and Central Finland. The secondary aim of this study was to evaluate the associations of CD4^+^ and FOXP3^+^ T cell infiltrations on patient survival. We also aimed to characterize the immune cell microenvironment in SB-NETs, SB-NET lymph node metastases and G-NETs.

## 2. Materials and Methods

### 2.1. Patients

In this study, patients diagnosed with histologically confirmed ileal or jejunal SB-NETs (*n* = 125) in Oulu University Hospital from 9 February 2000 to 7 February 2018 and in Central Finland Central Hospital from 24 February 2000 to 31 December 2017 were included. Of these patients, 99 had SB-NET lymph node metastases. All the patients diagnosed with histologically confirmed G-NETs of the corpus of the stomach (*n* = 26) in Oulu University Hospital from 9 February 2000 to 7 February 2018 were also included as a separate cohort and one of these patients had lymph node metastases. Clinical data were gathered from electronic patient records. Survival data were acquired from the Cause of Death Registry maintained by Statistics Finland. The end of follow-up was set to 31 December 2019. The study was approved by both involved hospital districts. The need for informed consent was waived and the use of data and samples approved by the National Authority for Medicolegal Affairs (VALVIRA) and by the Ethics Committee of Oulu University Hospital.

Patients with SB-NETs underwent right side hemicolectomy (*n* = 33), resection of the small intestine (*n* = 82) and tumor biopsy by endoscopy (*n* = 6) and by laparotomy (*n* = 1). A small number of patients (*n* = 3) underwent other operations, mainly palliative surgery. Patients with G-NETs had endoscopic mucosal resection (*n* = 7), endoscopic biopsy (*n* = 17), gastric resection (*n* = 1) and gastrotomy with tumor excision (*n* = 1). The histological diagnosis was confirmed by an expert pathologist at the time of diagnosis. Tumor stage was determined according to the 8th edition of the AJCC/UICC TNM categories [15]. Tumor grade was determined according to the 2019 WHO classification of tumors of the digestive system [16]. Median follow-up time in SB-NET patients was 5.8 (IQR 3.3–10.4) years.

### 2.2. Immunohistochemical Staining

Tissue samples from tumors and lymph node metastases were fixed in formalin and embedded in paraffin blocks at the time of diagnosis. In the preparation of microscopic slides, the sample blocks were retrieved from archives and cut into tissue sections of 3.5 µm in thickness. The sections were deparaffinized and rehydrated. Antigen retrieval was performed with tris-EDTA buffer at pH 9 in a microwave oven. Tissue sections were cooled at room temperature and rinsed. Endogenous peroxidase activity was neutralized in peroxidase blocking solution (Dako, Glostrup, Denmark, S2023). Tissue sections were incubated with mouse monoclonal antibodies. Anti-CD3 antibody concentrate (Novocastra, Newcastle, UK, NCL-L-CD3-565) was used in 1:50 dilution for 30 min, anti-CD8 (Novocastra, NCL-L-CD8-4B11) in 1:200 dilution for 30 min, anti-CD4 (Novocastra, NCL-L-CD4-368) in 1:100 dilution for 60 min and anti-FOXP3 (Abcam, Cambridge, UK, ab20034) in 1:100 dilution for 30 min. EnVision polymer (Dako, K5007) was used as a secondary antibody. Visualization of the staining was performed with diaminobenzidine working solution (Dako, K5007) and counterstaining was done with hematoxylin.

### 2.3. Scoring

Microscopic slides were scanned at ×20 magnification using an Aperio AT2 digital slide scanner. For summary of the study cohort and used methods, see Figure 1.

CD3^+^, CD8^+^, CD4^+^ and FOXP3^+^ immunohistochemistry was analyzed by J.P.V. using a previously validated cell counting method and QuPath v. 0.1.2 software (created at the Centre for Cancer Research & Cell Biology, Queen’s University Belfast, developed at the University of Edinburgh, Edinburgh, UK) [17,18]. Immune cell densities were measured at the tumor center and at the invasive margin. The invasive margin was defined as a region extending 0.5 mm from both sides of the tumor border (Figure 2).

To calculate ICS, dichotomization into groups of low and high was done for both CD3^+^ and CD8^+^ T cell densities. Medians were chosen as cut-off values. The cut-off values (cells/mm^2^) were 156 for CD3^+^ at the tumor center, 156 for CD3^+^ at the invasive margin, 67 for CD8^+^ at the tumor center and 62 for CD8^+^ at the invasive margin. The cut-off values for lymph node metastases were 202 for CD3^+^ at the tumor center, 269 for CD3^+^ at the invasive margin, 64 for CD8^+^ at the tumor center and 107 for CD8^+^ at the invasive margin. For each tumor, this yielded four dichotomized density variables. ICS from 0 to 4 was determined by using the sum of the four dichotomized density variables. For each sample, a density variable at or below the median awarded no points towards the total sum and each density variable above the median awarded one point.

The densities of CD4^+^ and FOXP3^+^ T cells were dichotomized in the fashion mentioned above. The cut-off values for SB-NETs were 48 for CD4^+^ at the tumor center, 60 for CD4^+^ at the invasive margin, 6.5 for FOXP3^+^ at the tumor center and 2.9 for FOXP3^+^ at the invasive margin. The cut-off values for lymph node metastases were 68 for CD4^+^ at the tumor center, 72 for CD4^+^ at the invasive margin, 8.3 for FOXP3^+^ at the tumor center and 12.5 for FOXP3^+^ at the invasive margin.

For G-NETs, the densities of CD3^+^, CD8^+^, CD4^+^ and FOXP3^+^ T cells were determined by counting the number of positive cells per mm^2^.

### 2.4. Statistical Analysis

Survival times were calculated from the date of surgery or biopsy until the time of death or the end of follow-up. Disease-specific survival (DSS) rates were calculated using the Kaplan-Meier method with a log-rank test stratified by ICS (low 0–1 and high 2–4). Crude and adjusted hazard ratios (HRs) for mortality were calculated using a Cox regression model. Cox regression was adjusted for age, sex, stage (I–II, III, IV), adjuvant somatostatin therapy (yes/no) and sample type (surgical resection specimen or biopsy). A *p*-value of less than 0.05 was considered significant. The statistical analyses were performed with IBM SPSS statistics 26 for Windows (IBM Corporation, Armonk, NY, USA).

## 3. Results

### 3.1. Patient Demographics

Of the whole cohort, including 125 SB-NETs and 26 G-NETs, 19 were excluded from the final analysis due to inadequate sample material, resulting in 113 SB-NETs and 19 G-NETs. A total of 92 from the 113 SB-NET patients were reported to have lymph node metastases. Tissue samples of lymph node metastases were available from 75 of the 92 patients and one sample per patent was acquired for this research.

Of the 113 SB-NET patients, 13 (11.5%) were stage I–II, 60 (53.1%) were stage III and 40 (35.4%) were stage IV. Baseline characteristics are shown in Table 1. Median follow-up time was 5.9 years (IQR 3.3–10.4). Forty-four patients died during follow-up, 26 of these deaths were disease-specific.

G-NETs were excluded from survival analyses and only used to compare immune cell densities between SB-NETs and G-NETs. Baseline characteristics are listed in Table 1.

### 3.2. Immune Cell Score—No Prognostic Value in Primary Tumors or Lymph Node Metastases

In SB-NETs, ICS was low in 40.7% of cases and high in 59.3%. No significant difference in DSS was observed between low and high ICS groups at 5 years (92.6% vs. 87.8%) or 10 years (53.8% vs. 79.4%) (Figure 3a, Table 2, *p* = 0.507). Similarly, in SB-NET lymph node metastases, ICS was low in 41.3% of cases and high in 58.7%. ICS in lymph node metastases was not associated with survival (Figure 3b, Table 2). Some differences in absolute survival percentages were seen, but the survival curves overlapped and crossed throughout the follow-up (Figure 3a,b).

Crude and adjusted HRs for disease-specific mortality are shown in Table 3. The crude HR in SB-NETs in the high ICS group was 0.75 (95% CI 0.32–1.75) and the adjusted HR was 1.37 (95% CI 0.56–3.38). In SB-NET lymph node metastases, the crude HR in the high ICS group was 1.56 (95% CI 0.47–5.21) and the adjusted HR was 1.51 (95% CI 0.39–5.81).

### 3.3. Individual Lymphocyte Densities in Primary Tumors Have No Prognostic Value

No significant difference was found in DSS between the low and high inflammation groups based on individual dichotomized densities of the studied cell types (Table 4). The results were conclusively negative both at the tumor center and at the invasive margin in all T cell groups. DSS rates for each cell type are presented in Table 4.

### 3.4. Individual Lymphocyte Densities in Lymph Node Metastases Have No Prognostic Value

DSS rates in lymph node metastases are shown in Table 5. Some differences in absolute survival percentages are seen but the survival overlapped during most of the follow-up. No statistical significance between low and high cell densities was seen.

### 3.5. Immune Microenvironment in SB-NETs and G-NETs—Lower T Cell Densities in SB-NETs

SB-NETs had fewer CD3^+^ and CD8^+^ T cells at the tumor center (Figure 4a) and at the invasive margin (Figure 4b) than G-NETs. In SB-NETs and SB-NET lymph node metastases, the T cell profiles were nearly identical (Figure 5a,b).

## 4. Discussion

In our study, we found that densities of CD3^+^, CD8^+^, CD4^+^ and FOXP3^+^ T cells in SB-NETs and SB-NET lymph node metastases were not significantly associated with DSS. Primary tumors and lymph node metastases seemed to have similar T cell densities, whereas G-NETs had higher densities of T cells.

The findings of our current study seem to be in line with some previous research. A study by da Silva et al., involving 64 SB-NETs, characterized the intratumoral and extratumoral CD3^+^ and CD8^+^ T cell densities by manual cell counting [19]. They used the average number of T cells per high power field to categorize the cell densities in groups of low (0–20 cells/field) and high (>20 cells/field) densities. They found low intratumoral densities of CD3^+^ cells in 94% of cases and high densities in 4%. Intratumoral CD8^+^ cell density was low in 100% of cases. That study reported no clear association between T cell densities and survival. A study by Xing et al. analyzed the inflammation in 33 neuroendocrine carcinomas of the digestive tract by semiquantitative scores based on percentages of staining cell infiltration of the stromal area (low infiltration was 0–25% staining and high infiltration >25% staining) [20]. They found intratumoral CD3^+^ T cell infiltration in 69.7% of patients and CD8^+^ T cell infiltration in 27.3%. They found no difference in overall survival between different densities of CD3^+^ or CD8^+^ T cells. Contrary to our finding, Xing et al. found some tumors lacking T cell infiltrates [20]. This may be due to underlying biological differences between neuroendocrine carcinomas and NETs. Ferrata et al. investigated 22 samples of neuroendocrine neoplasms with a high proliferation index (Ki-67 > 20%) and characterized the intratumoral infiltration of CD3^+^ and CD8^+^ T cells by progressive classification: 0 (no staining), 1+ (weak staining), 2+ (moderate staining) and 3+ (strong staining) [21]. They found a larger number of patients (45.5%) showing increased CD3^+^ T cell densities, whereas fewer patients (18.2%) showed increased CD8^+^ T cell densities. Survival analyses were not performed. A study by Lamarca et al. reported a significant presence of CD3^+^ and CD8^+^ T cells in 62 G1–2 SB-NETs [22]. They evaluated cell densities by a semiquantitative score: none (no immune infiltrates), focal (mostly perivascular infiltrate with some intratumoral extension), moderate (prominent extension of immune infiltrates away from perivascular areas and amongst tumor cells) and severe (immune infiltrates obscuring the tumor). Focal CD3^+^ T cell infiltration was observed in 88.6% of patients and moderate infiltration in 7.1%. Focal CD8^+^ T cell infiltration was observed in 92.6% of patients and moderate infiltration in 4.3%. Survival analysis was inconclusive due to low power. Compared to these previous studies, our study benefits from a quantitative assessment of T cell infiltrates using a continuous scale, as well as stronger statistical power in survival analysis, related to a larger sample size.

The presence and characteristics of T helper populations may also harbor significance. Lamarca et al. investigated CD4^+^ T cells in 62 G1–2 SB-NETs [22]. They observed focal CD4^+^ T cell infiltration in 42.9% of patients and moderate infiltration in 2.9%, but no link between CD4^+^ T cell infiltration and survival could be observed due to limited power. Da Silva et al. described FOXP3^+^ T cell densities in 64 SB-NETs [19]. They found no clear association between FOXP3^+^ T cell densities and survival.

Some studies evaluating pNETs have provided opposing results compared with our findings. Katz et al. examined immune cell infiltrations in 87 pNETs and found that CD3^+^ T cell infiltration was associated with improved recurrence-free survival [23]. Wei et al. calculated the Immunoscore in 158 pNETs and proposed the Immunoscore system for pNETs (ISpnet) as a predictive tool to assess prognosis [11]. In their study, low peritumoral CD4^+^ cell infiltration, high intratumoral CD8^+^ cell infiltration and low peritumoral CD8^+^ cell infiltration were significantly associated with recurrence-free survival. Katz et al. investigated 39 NET liver metastases [23]. They evaluated lymphocyte densities by manual cell counting per 10 high power fields and scored the samples from 0 to 3 based on predetermined increments. Cut-off points for analyses (low vs. high cell densities) were assigned based on median score. They found that a low level of FOXP3^+^ lymphocytes in NET liver metastases was a predictor for prolonged overall survival [23]. De Reuver et al. found increasing FOXP3^+^ T cell densities to be associated with worse disease-specific survival in a cohort of 101 pNETs [24]. For analyses, they categorized the cases in grades from 0 (no lymphocytes) to 2 (five or more lymphocytes) based on the amount of positively stained FOXP3^+^ T cells in a one-millimeter diameter core.

These studies involving positive association with T cell densities and survival were conducted in pNETs. The tumor biology in SB-NETs is thought to differ from that of pNETs. SB-NETs are generally seen as mutationally silent tumors and they bear a lower mutational burden than pNETs [19,25,26]. High mutational burden is often associated with a stronger host immune response since they are “altered self” and immune tolerance has not developed [11,19,23,24,27]. A recent study found driver mutations (27 mutations in total, including TP53, RB1, CDKN1B, KRAS and NRAS mutations) occurring in 50% of metastasized SB-NETs but no difference in survival was seen in patients with or without these mutations [28].

Our present study has several strengths. This study involved 113 SB-NETs, representing one of the largest published cohorts evaluating immune cells in these tumors. For the first time, we included 75 SB-NET lymph node metastases to compare immune cell infiltrates in primary NETs and nodal metastases and assess whether immune cells evaluated in nodal disease could predict progression of SB-NETs. We used whole section slides to evaluate T cell infiltrates whereas many previous studies investigating tumor microenvironment have used tissue microarray slides. Our study included a long follow-up time, which is particularly important in often slowly progressing diseases like SB-NETs. Furthermore, based on reliable registry data from Statistics Finland, causes of death were available, making DSS analyses possible. Still, our study has some limitations. SB-NETs are rare and despite including patients from a period of almost two decades, the sample size was relatively small and strong conclusions regarding our negative result cannot be made, and replication studies are needed. Due to the long time period, the treatment and diagnostics of these patients have possibly improved and this could cause some confounding. However, the treatment approach to SB-NETs has remained rather stable and this confounding is probably small. The study population consisted only of G1 and G2 well-differentiated NETs and the information produced cannot be applied to G3 NETs or neuroendocrine carcinomas. Some studies involving NETs and TILs report the programmed death-ligand 1 (PD-L1) status of the tumors and characterize the immune microenvironment by combining this information of T cell infiltrates and PD-L1 [29]. The combined PD-L1-TIL status has been used to predict response to treatment with immune checkpoint inhibitors in other cancers [21,30]. PD-L1 status was not included in our study and this can be seen as a limitation. Lamarca et al. found tumoral PD-L1 expression to have no impact on overall survival, recurrence-free survival or progression-free survival in well-differentiated SB-NETs [22]. Their results were similar for the expression of PD-1 in TILs. Other immune checkpoint ligand molecules, such as CTLA-4, galectin-9 and CD155, have been investigated in some neuroendocrine tumors and their expression has been associated with better prognosis and response to immune checkpoint inhibitors [31,32,33]. The characterization and role of these ligand molecules in SB-NETs require further research. The number of G-NETs (*n* = 19) was rather small, and this may reduce the applicability of the results in describing the G-NET immune microenvironment. Of the 19 G-NET patients, 15 had atrophic gastritis. The evaluation of TILs in G-NETs may be confounded by the naturally occurring inflammation in atrophic gastritis. However, G-NETs were not used in survival analyses, but only to compare T cell densities in NETs of different sites in the gastrointestinal tract.

The current study has some potential clinical implications. The immune microenvironment is being increasingly studied and even proposed as an additional prognostic tool to complement traditional TNM staging [9]. Our study highlights heterogeneity between different cancers, suggesting that host immune reaction has tumor type-specific prognostic significance. Further studies are needed to confirm our findings.

## 5. Conclusions

In our study, we reported the presence of CD3^+^, CD4^+^, CD8^+^ and FOXP3^+^ T cells in SB-NETs. No association with ICS or T cell densities and prognosis was seen in SB-NETs. Cell densities were similar in primary SB-NETs and their lymph node metastases, but higher in G-NETs. For future research, replication studies are needed to confirm our findings. Studies with larger study populations and the inclusion of immune checkpoint ligand molecules are warranted. All this could provide additional information about survival and the applicability of novel immunological anticancer therapies in SB-NET patients.

## Figures and Tables

**Figure 1 diseases-09-00042-f001:**
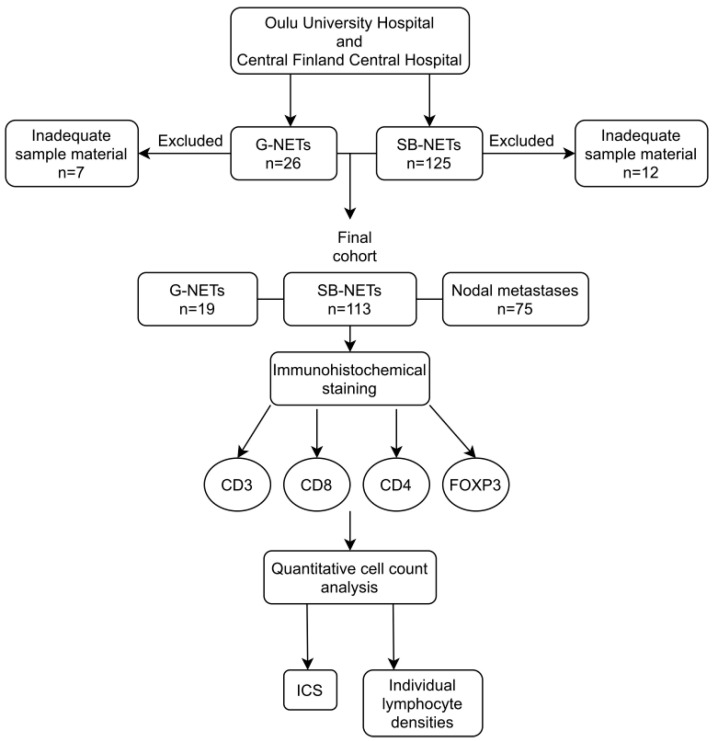
A flowchart summarizing the study cohort and the used methods.

**Figure 2 diseases-09-00042-f002:**
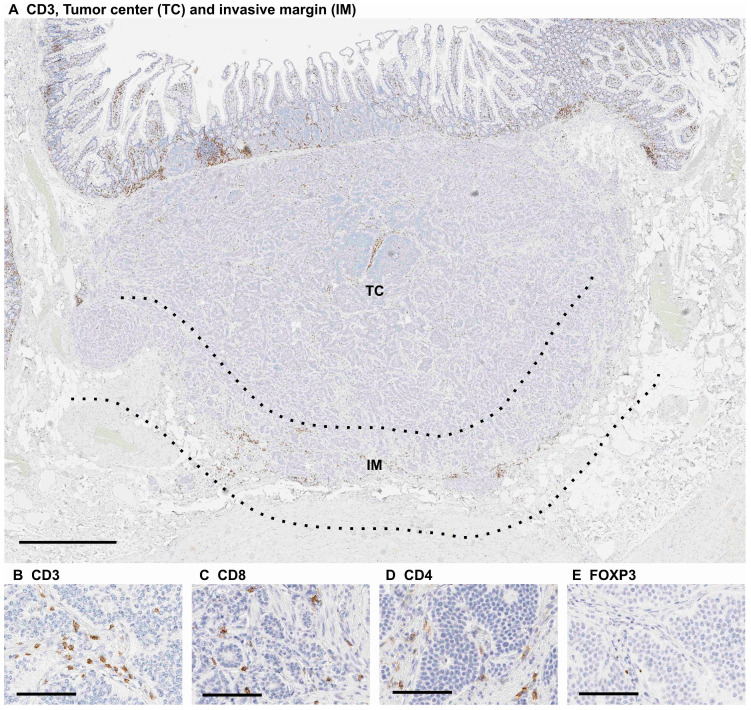
(**A**) CD3^+^ immunohistochemistry in small bowel neuroendocrine tumor tissue showing the tumor center (TC) and the invasive margin (IM) used to determine immune cell score: (**B**–**E**) Examples of CD3^+^, CD8^+^, CD4^+^ and FOXP3^+^ T cell infiltrations in immunohistochemical stainings. Scale bars are 1 mm (**A**) and 100 µm (**B**–**E**).

**Figure 3 diseases-09-00042-f003:**
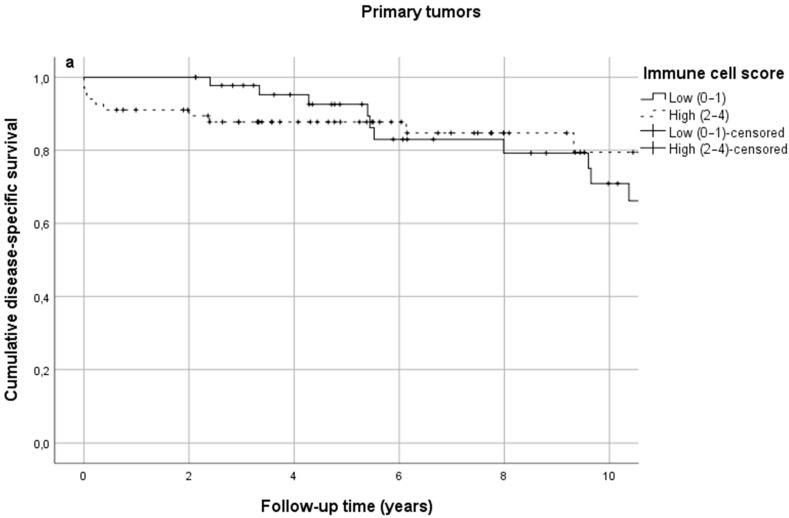
(**a**) Cumulative disease-specific survival in primary small bowel neuroendocrine tumors stratified by immune cell score; (**b**) cumulative disease-specific survival in small bowel neuroendocrine tumor lymph node metastases stratified by immune cell score.

**Figure 4 diseases-09-00042-f004:**
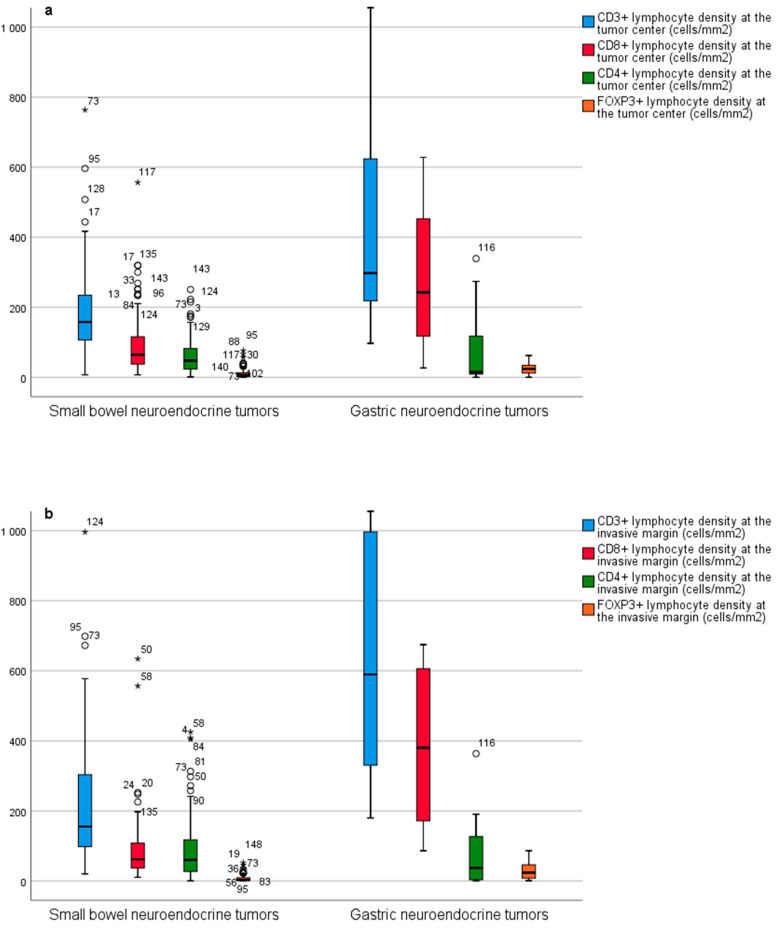
(**a**) Comparison of CD3^+^, CD8^+^, CD4^+^ and FOXP3^+^ immune cell densities between primary small bowel neuroendocrine tumors and primary gastric neuroendocrine tumors at the tumor center and (**b**) at the invasive margin. Outliers are marked with asterisks (*).

**Figure 5 diseases-09-00042-f005:**
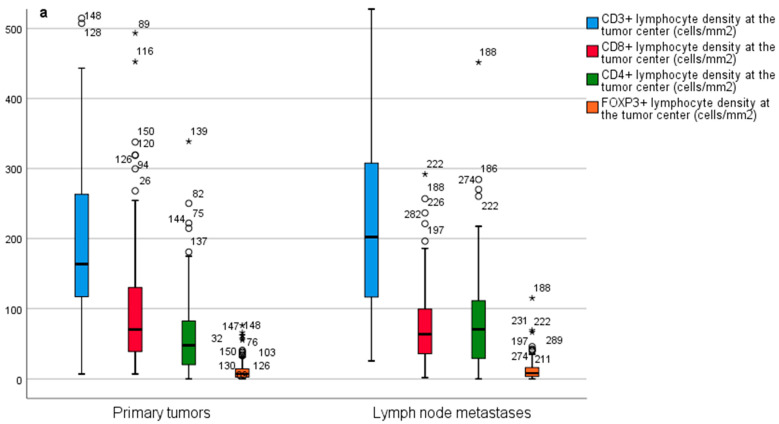
(**a**) Comparison of CD3^+^, CD8^+^, CD4^+^ and FOXP3^+^ immune cell densities between primary small bowel neuroendocrine tumors and small bowel neuroendocrine tumor lymph node metastases at the tumor center and (**b**) at the invasive margin. Outliers are marked with asterisks (*).

**Table 1 diseases-09-00042-t001:** Baseline characteristics of the study population.

Variables	Small Bowel NET, *n* = 113	Gastric NET, *n* = 19
Sex		
Female, *n* (%)	50 (44.2)	12 (63.2)
Male, *n* (%)	63 (55.8)	7 (36.8)
Age, median (IQR) years	65.2 (55.3–72.4)	61.7 (52.8–68.1)
**Immune Cell Score, Primary Tumors**		
0, *n* (%)	27 (23.9)	
1, *n* (%)	19 (16.8)	
2, *n* (%)	26 (23.0)	
3, *n* (%)	14 (12.4)	
4, *n* (%)	27 (23.9)	
**Immune Cell Score, Lymph Node Metastases**		
0, *n* (%)	18 (24.0)	
1, *n* (%)	13 (17.3)	
2, *n* (%)	12 (16.0)	
3, *n* (%)	17 (22.7)	
4, *n* (%)	15 (20.0)	
**T-Class**		
TX, *n* (%)	6 (5.3)	11 (57.9)
Tis, *n* (%)	0	1 (5.3)
T1, *n* (%)	5 (4.4)	3 (15.8)
T2, *n* (%)	23 (20.4)	4 (21.1)
T3, *n* (%)	43 (38.1)	0
T4, *n* (%)	36 (31.9)	0
**N-Class**		
N0, *n* (%)	21 (18.6)	18 (94.7)
N1–2, *n* (%)	92 (81.4)	1 (5.3)
**M-Class**		
M0, *n* (%)	73 (64.6)	18 (94.7)
M1, *n* (%)	40 (35.4)	1 (5.3)
**Stage**		
I-II, *n* (%)	13 (11.5)	18 (94.7)
III, *n* (%)	60 (53.1)	0
IV, *n* (%)	40 (35.4)	1 (5.3)
**Grade**		
Not available, *n* (%)	2 (1.8)	3 (15.8)
1, *n* (%)	87 (77.0)	9 (47.4)
2, *n* (%)	24 (21.2)	7 (36.8)
3, *n* (%)	0	0
**Tumor Location**		
Ileum, *n* (%)	106 (93.8)	
Jejunum, *n* (%)	7 (6.2)	
Gastric corpus, *n* (%)		19 (100)
**Somatostatin Analog Treatment**		
Yes, *n* (%)	62 (54.9)	1 (5.3)
**Chemotherapy**		
No, *n* (%)	96 (85.0)	19 (100)
Preoperative, *n* (%)	5 (4.4)	0
Postoperative, *n* (%)	12 (10.6)	0
**Multiple Primary Tumors**		
Yes, *n* (%)	32 (28.3)	11 (57.9)
**P-CgA**		
≥3 nmol/L, *n* (%)	87 (77.0)	10 (52.6)
Median (IQR) nmol/L	5.8 (3.4–14.0)	9.4 (5.1–17.9)
**dU-5-HIAA**		
≥42 umol/L, *n* (%)	52 (46.0)	1 (5.3)
Median (IQR) umol/l	44.0 (21.0–133)	25.0 (17.5–47.0)
**Atrophic Gastritis, *n* (%)**	0	15 (78.9)

P-CgA, plasma chromogranin A; dU-5-HIAA, 24 h urine hydroxyindoleacetic acid.

**Table 2 diseases-09-00042-t002:** Disease-specific survival (DSS) rates based on immune cell score (ICS) in primary small bowel neuroendocrine tumors and lymph node metastases.

DSS	No. of Patients	ICS Low (0–1)	ICS High (2–4)	*p*
Primary tumors	113	5-year: 92.6%	5-year: 87.8%	
		10-year: 53.8%	10-year: 79.4%	0.507
Lymph node metastases	75	5-year: 88.9%	5-year: 90.4%	
		10-year: 71.1%	10-year: 59.8%	0.466

**Table 3 diseases-09-00042-t003:** Hazard ratios (HRs) for disease-specific mortality with 95% confidence intervals (CIs) in primary small bowel neuroendocrine tumors and lymph node metastases in low and high immune cell score groups.

	No. of Patients	ICS Low (0–1)	ICS High (2–4)
**Primary Tumors**			
Crude	113	1.00 (reference)	0.75 (CI 0.32–1.75)
Adjusted	113	1.00 (reference)	1.37 (CI 0.56–3.38)
**Lymph Node Metastases**			
Crude	75	1.00 (reference)	1.56 (CI 0.47–5.21)
Adjusted	75	1.00 (reference)	1.51 (CI 0.39–5.81)

Adjusted for age, sex, stage (I–II, III, IV), adjuvant somatostatin therapy (yes/no) and sample type (surgical resection specimen or biopsy).

**Table 4 diseases-09-00042-t004:** Disease specific-survival (DSS) rates according to dichotomized (based on median values) CD3^+^, CD8^+^, CD4^+^ and FOXP3^+^ T cell densities at the tumor center (TC) and at the invasive margin (IM) in primary small bowel neuroendocrine tumors.

DSS		TC Low	TC High	*p*-Value	IM Low	IM High	*p*-Value
CD3^+^	5-year	90.00%	89.10%		88.50%	90.80%	
	10-year	54.80%	77.60%	0.584	60.50%	68.10%	0.612
CD8^+^	5-year	90.40%	90.90%		92.40%	88.60%	
	10-year	63.70%	69.10%	0.762	58.90%	80.50%	0.547
CD4^+^	5-year	91.90%	90.80%		90.20%	92.60%	
	10-year	63.50%	71.70%	0.913	64.40%	66.70%	0.884
FOXP3^+^	5-year	89.80%	91.00%		90.30%	90.80%	
	10-year	61.70%	70.10%	0.901	60.40%	73.00%	0.832

**Table 5 diseases-09-00042-t005:** Disease-specific survival (DSS) rates according to dichotomized (based on median values) CD3^+^, CD8^+^, CD4^+^ and FOXP3^+^ T cell densities at the tumor center (TC) and at the invasive margin (IM) in small bowel neuroendocrine tumor lymph node metastases.

DSS		TC Low	TC High	*p*-Value	IM Low	IM High	*p*-Value
CD3^+^	5-year	90.50%	91.60%		85.10%	97.30%	
	10-year	72.40%	57.70%	0.366	70.90%	60.90%	0.798
CD8^+^	5-year	91.10%	88.40%		90.90%	88.90%	
	10-year	71.90%	53.00%	0.337	68.70%	61.60%	0.516
CD4^+^	5-year	88.00%	91.60%		88.00%	91.90%	
	10-year	51.90%	77.20%	0.539	63.40%	65.30%	0.737
FOXP3^+^	5-year	87.90%	91.70%		85.60%	94.40%	
	10-year	75.30%	49.70%	0.335	67.60%	51.60%	0.832

## Data Availability

Anonymized data are available upon request from the corresponding author.

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
