# Peer review of "CD3+, CD8+, CD4+ and FOXP3+ T Cells in the Immune Microenvironment of Small Bowel Neuroendocrine Tumors"

_diseases, 2021, doi:10.3390/diseases9020042_

Round 1

Reviewer 1 Report

  1. Line 104: should read Diaminobenzidine
  2. "The final cohort included 75patients SB-NET lymph node metastases." - this is not clear, does it mean 75 lymph nodes or 75 patients with metastases?
  3. How was statistical significance calculated for table 2? 10y survival for primary tumours seem to be significantly different between ICS low and high groups.
  4. Please move strengths/weaknesses towards the end of the discussion
  5. Please slightly rewrite the results and discussion for increased readability e.g. consider changing the subheadings in the results into more descriptive ones like X was increased due to Y.

Overall, it is an interesting study with relatively high number of analysed cases and certain value.

Reviewer 2 Report

I read with interest the Manuscript titled "CD3+, CD8+, CD4+ and FOXP3+ T cells in the immune microenvironment of small bowel neuroendocrine tumors". The topic is really interesting and attracts the reader's attention. The conception of the study is adequate, the introduction contains enough information to understand the background of the theme.

The manuscript provides data about the possible prognostic significance of immune cell score (ICS) based on CD3+ and CD8+ T cells in a retrospective, consecutive series of SB-NETs. The authors characterized the immune cell microenvironment in SB-NETs, SB-NET lymph node metastases, and G-NETs using the adequate scoring technique.

This manuscript is worth publication, however, the following questions need to be discussed:

In Table 1. I do not see data about female patients. Did you involve only male patients in the study?

Indeed the evaluation of the PD-L1 status of the tumor cells certainly adds important data about the immune microenvironment and possible association with survival. Besides PD-L1 could other immune checkpoint ligand molecules (CTLA-4, Galectin-9, CD155) be an important factor in tumor growth?

Also, the PD-1 expression levels of the tumor-infiltrating lymphocytes together with the PD-L1 status of the tumor cells might have a prognostic value about the survival. Are there any studies available regarding this information?

Reviewer 3 Report

In this work, the role of the several T cell types in the immune u-environment of small bowel neuroendocrine tumours was studied.

The SB-NET lymph node metastases and gastric neuroendocrine tumours was subjected to evaluate prognostics of immune cell infiltrates.

The work is exhaustively presented and easy-to-comprehend to the reader.

Regarding English, only small check should be performed.

The length is acceptable.

I have only several remarks to add that I consider vital to add.

REMARKS

INTRODUCTION (note)

The studies of disease microenvironments and adjacent sections on the molecular basis is a hot topic today. This is for sure. Many researchers struggle to downscale perimeter-of-interest to describe the actual “health” status of small number of regions consisted of several hundreds of cells or even to study biological section on the level of single cells or even sub-cellular levels – i.e., actual molecular status of cell organelles. Thus I consider important to add some statement of where the current study of microenvironments is heading to. Please, see the down below upgrade.

INTRODUCTION (upgrade, last paragraph)

Contrary to many other tumor types, relatively few studies have evaluated the SB-60 NET immune microenvironment. We see that trend to evaluate disease microenvironments is significantly growing [https://doi.org/10.1002/pmic.202000318, https://doi.org/10.1016/j.bbapap.2021.140658]. Therefore, the primary aim of this study was to test the prognostic significance of immune cell score (ICS) based on CD3+ and CD8+ T cells in a retrospective, consecutive series of microenvironment SB-NETs from two institutions in Northern and Central Finland.

MATERIALS AND METHODS (note)

Please, draw an analytical workflow, to provide a reader a quick glance on the suggested work.

CONCLUSION (note)

In conclusion, please provide in several points your future plans in this scope of research.

Round 2

Reviewer 3 Report

Authors have accomplished all given remarks. Thus I consider its publication at current form.

Author Response

We thank the Reviewer for their comment.